# A Biologically Plausible Dense Associative Memory with Exponential Capacity

**Mohadeseh Shafiei Kafraj**
Gatsby Computational Neuroscience Unit
University College London
London W1T 4JG, UK
mohadeseh.kafraj.22@ucl.ac.uk

**Dmitry Krotov**
IBM Research
krotov.a.dmitry@gmail.com

**Peter E. Latham**
Gatsby Computational Neuroscience Unit
University College London
London W1T 4JG, UK
pel@gatsby.ucl.ac.uk

## Abstract

Krotov and Hopfield (2021) proposed a biologically plausible two-layer associative memory network with memory storage capacity exponential in the number of visible neurons. However, the capacity was only linear in the number of hidden neurons. This limitation arose from the choice of nonlinearity between the visible and hidden units, which enforced winner-take-all dynamics in the hidden layer, thereby restricting each hidden unit to encode only a single memory. We overcome this limitation by introducing a novel associative memory network with a threshold nonlinearity that enables distributed representations. In contrast to winner-take-all dynamics, where each hidden neuron is tied to an entire memory, our network allows hidden neurons to encode basic components shared across many memories. Consequently, complex patterns are represented through combinations of hidden neurons. These representations reduce redundancy and allow many correlated memories to be stored compositionally. Thus, we achieve much higher capacity: exponential in the number of hidden units, provided the number of visible units is sufficiently large relative to the number of hidden units. Exponential capacity arises because all binary states of the hidden units can become stable memory patterns. Moreover, the distributed hidden representation, which has much lower dimensionality than the visible layer, preserves class-discriminative structure, supporting efficient nonlinear decoding. These results establish a new regime for associative memory, enabling high-capacity, robust, and scalable architectures consistent with biological constraints.

## 1 Introduction

Associative memory networks are a class of attractor models in which the system can recall stored memories from their incomplete or noisy versions via recurrent dynamics (Krotov et al. (2025)). In such models, memories are conceptualized as the stable fixed points of the network dynamics. The number of fixed points determines the storage capacity of the network, and significant effort has been made to construct networks with sufficiently high storage capacity to explain human memory.

The classical Hopfield network, a leading model for associative memory, has a storage capacity that scales linearly with the number of neurons in the network (Hopfield (1982)). Dense associative memories, sometimes also referred to as modern Hopfield networks (Krotov and Hopfield (2016)), are promising modifications of the classical Hopfield model. By incorporating higher-order interactions (e.g., interactions that are quadratic, rather than linear, in the input to a neuron), they achieve a storage capacity that scales super-linearly with the number of neurons. There are many possible choices for the energy function in this class of models. For instance, the power interaction vertex leads to the

power-law scaling of the capacity (Baldi and Venkatesh (1987); Gardner (1987); Abbott and Arian (1987); Horn and Usher (1988); Chen et al. (1986); Krotov and Hopfield (2016)). More sophisticated shapes of the energy function result in exponential storage capacity in the number of neurons, while maintaining large basins of attraction (Demircigil et al. (2017); Lucibello and Mézard (2024)).

The naive implementation of Dense Associative Memory models, however, relies on synaptic interactions that are challenging to implement in biological circuits. In particular, these models require nonlinear interactions among synapses. While several biological mechanisms could in principle support restricted forms of higher-order interactions, such as astrocytic processes, dendritic computations, or distributed neurotransmitter effects, these remain limited in scope and dictate strong constraints on the possible shape of the energy function (Burns and Fukai (2023); Kozachkov et al. (2025); Kafraj et al. (2025)). The implementation of Dense Associative Memory introduced by Krotov and Hopfield (2021) does not suffer from these limitations, as it relies only on standard synaptic interactions. In this architecture, the visible neurons correspond to features of the patterns, while the hidden neurons serve as auxiliary computational elements that mediate complex interactions. Higher-order interactions among visible neurons emerge by selecting appropriate activation functions for the hidden neurons.

Nevertheless, the implementation of Krotov and Hopfield has two key limitations. First, the storage capacity is at most linear in the number of hidden neurons (Krotov (2021); Krotov and Hopfield (2021)). This is unsatisfactory from the perspective of information storage – one would like to store as much information as possible while utilizing only a small number of neurons. Second, at inference time, the network demonstrates a winner-take-all behavior. This means that the asymptotic fixed point that the network converges to corresponds to a single hidden neuron being activated, while the rest of the hidden neurons are inactive. This behavior results in grandmother-like representations for hidden neurons, as opposed to distributed representations, which are more efficient at storing information.

Our work tackles these two limitations. Specifically, we present a novel implementation of Dense Associative Memory that achieves exponential storage capacity in the number of hidden neurons. This is accomplished with a simple yet critical change: we use a threshold activation function that does not enforce winner-take-all dynamics. The threshold activation enables distributed memory representations–multiple hidden neurons can be active for a memory, and each hidden neuron can participate in multiple memories. As a result, all possible binary patterns of hidden neuron states become stable fixed points, enabling the network to store exponentially many memories, including highly correlated ones. Beyond high capacity, the hidden layer of the network is low-dimensional compared to the visible layer, yet it produces structured representations that preserve class-discriminative information, with memories sharing components represented close together in the hidden activity space. We establish this result through both theoretical analysis and numerical simulations, and show that the resulting fixed points also possess large basins of attraction.

Our model is closely related to the framework recently proposed by Chandra et al. (2025), which combines multiple Dense Associative Memory modules to produce a distributed code for the visible neurons. In that work, each module performs a winner-take-all operation similar to Krotov and Hopfield (2021), so only a single hidden neuron is active per module. By combining several modules, though, they achieve exponential storage capacity. However, we show that multiple modules are unnecessary: exponential capacity can be achieved with a single module, provided the activation function is chosen appropriately.

Beyond its biological motivation, our work also connects to a growing body of research on Dense Associative Memories in machine learning. Notably, it has been shown that Dense Associative Memory closely corresponds to the attention mechanism in transformer architectures (Ramsauer et al. (2021); Hoover et al. (2023a)), offering a principled framework for viewing the transformer's attention and feedforward computations as steps in a global energy minimization process. Complementary research has demonstrated that generative diffusion models, widely used in high-quality image generation, also exhibit associative memory behavior (Hoover et al. (2023b); Ambrogioni (2024); Pham et al. (2025)). Further studies have expanded the model's functionality: for instance, Chaudhry et al. (2023) examined its ability to store and retrieve long sequences; Dohmatob (2023) and Hoover et al. (2025) proposed alternative energy functions that also support exponential storage capacity; and Burns and Fukai (2023) introduced higher-order simplicial interactions. Our results contribute to this line of work by showing how exponential storage capacity can be achieved within a biologically plausible two-layer framework, thereby bridging theoretical neuroscience with modern machine learning architectures.

In the following sections, we first formally define the model and its dynamics and derive the optimal threshold analytically for a network with fixed weights. We then present a theoretical analysis of storage capacity and basins of attraction, showing that the network exhibits large basins of attraction, making recall robust to substantial noise in the visible inputs. Next, we introduce a learning rule for storing real, correlated memories, enabling compositional memory storage, and present numerical experiments on MNIST and CIFAR-10 that demonstrate high-capacity recall, structured hidden representations, and robustness to noise. Finally, we conclude by discussing the biological plausibility of the network and potential directions for extending the model to incorporate additional constraints and more realistic neuronal properties.

## 2 MODEL

In this section we present our model and demonstrate that its storage capacity scales exponentially with the number of hidden neurons, meaning that all possible binary patterns of hidden neurons are stable fixed points.

We first define the dynamics of the system as follows,

$$\tau_v \frac{dv_i}{dt} = -v_i + \frac{1}{\sqrt{N_h}} \sum_{\mu=1}^{N_h} \xi_{i\mu} \Theta(h_\mu - \theta) \tag{1a}$$

$$\tau_h \frac{dh_\mu}{dt} = -h_\mu + \frac{\sqrt{N_h}}{N_v} \sum_{i=1}^{N_v} \xi_{\mu i} v_i \,, \tag{1b}$$

where $\Theta(\cdot)$ is the standard Heaviside step function,

$$\Theta(z) = \begin{cases} 0 & \text{if } z \le 0 \\ 1 & \text{if } z > 0 \,. \end{cases} \tag{2}$$

The parameter $\theta$ will be chosen to ensure the stability of all binary patterns in the hidden layer.

The network consists of $N_v$ visible neurons (the $v_i$) and $N_h$ hidden neurons (the $h_\mu$), arranged in a bipartite architecture, i.e., without lateral connections within either layer.

Synaptic connections between visible neuron $i$ and hidden neuron $\mu$ are symmetric and drawn randomly from a standard normal distribution,

$$\xi_{\mu i} = \xi_{i\mu} \sim \mathcal{N}(0, 1). \tag{3}$$

The scaling factors in front of the sums in Eq. (1) are chosen purely for convenience, as they simplify subsequent expressions. Additionally, for simplicity, our theoretical analysis and experiments assume a Heaviside step function; however, Appendix F shows that this assumption is not required.

### 2.1 STORAGE CAPACITY

To determine the storage capacity, we'll first focus on the fixed points of the dynamics given in Eq. (1). Defining

$$s_\mu \equiv \Theta(h_\mu - \theta) \,, \tag{4}$$

it is straightforward to show that in steady state, $s_\mu$ satisfies

$$s_\mu = \Theta \left( \sum_{\nu=1}^{N_h} J_{\mu\nu} s_\nu - \theta \right) \tag{5}$$

where

$$J_{\mu\nu} \equiv \frac{1}{N_v} \sum_{i=1}^{N_v} \xi_{\mu i} \xi_{i\nu} \,. \tag{6}$$

Equation (5), with the weight matrix given in Eq. (6), is very close to the classical Hopfield model; the only difference is that in the classical model, the $\xi_{\mu i}$ are binary, whereas in our model they're Gaussian. However, the classical Hopfield model works in the regime $N_v < N_h$, with memory storage possible only if $N_v < 0.138 N_h$ (Amit et al. (1985)). Here, though, we'll consider a very different regime: $N_v \gg N_h$. In this limit, $J_{\mu\nu}$ approaches the identity matrix (Marchenko and Pastur (1967)), which decouples the hidden neurons. Assuming the threshold, $\theta$, is chosen correctly, this leads immediately to exponential storage capacity.

Exponential capacity clearly holds in the limit $N_v \to \infty$. What happens when $N_v$ is finite? We show in Appendix A.1 that

$$J_{\mu\nu} = \delta_{\mu\nu} + \frac{\zeta_{\mu\nu}}{\sqrt{N_v}} \tag{7a}$$

$$\zeta_{\mu\nu} \sim \mathcal{N}(0, 1 + \delta_{\mu\nu}). \tag{7b}$$

Here and in what follows $\delta_{\mu\nu}$ is the Kronecker delta: it's 1 if $\mu = \nu$ and 0 otherwise. Thus, Eq. (5) can be written as

$$s_\mu = \Theta\left(s_\mu + \frac{1}{\sqrt{N_v}} \sum_{\nu=1}^{N_h} \zeta_{\mu\nu} s_\nu - \theta\right). \tag{8}$$

Because the $\zeta_{\mu\nu}$ are independent, the second term in parentheses, which we denote $q_\mu$, scales as (see Eq. (24) in Appendix A.1)

$$|q_\mu| \equiv \left| \frac{1}{\sqrt{N_v}} \sum_{\nu=1}^{N_h} \zeta_{\mu\nu} s_\nu \right| \sim \sqrt{\frac{1}{N_v} \sum_{\nu=1}^{N_h} s_\nu^2 (1 + \delta_{\mu\nu})} \leq \sqrt{\frac{N_h + 1}{N_v}} \tag{9}$$

where the second inequality follows because $s_\nu$ is either 0 or 1.

If we set $\theta = 1/2$, in the limit $N_v \gg N_h$ Eq. (8) typically has two solutions: one at $s_\mu = 0$ and one at $s_\mu = 1$. However, if $q_\mu > 1/2$ then the only solution is $s_\mu = 1$, and if $q_\mu < -1/2$ then the only solution is $s_\mu = 0$. In both cases there can be a "bit flip", where $s_\mu$ is forced to the wrong value. In Appendix D (see in particular Eqs. (40) and (36), but with $\sigma_v = 0$) we show that

$$P_{\text{no bit flips}} \geq 1 - N_h \sqrt{\frac{N_h + 1}{N_v}} \frac{e^{-\frac{N_v}{8(N_h+1)}}}{\sqrt{\pi/2}}. \tag{10}$$

This bound approaches 1 exponentially rapidly as the number of visible units, $N_v$, increases.

There are exponentially many fixed points, but are they stable? To answer that, we need to do stability analysis. Combining Eq. (1) with the definitions of $s_\mu$, Eq. (4), and $J_{\mu\nu}$, Eq. (6), we have

$$h_\mu = \sum_\nu J_{\mu\nu} s_\nu. \tag{11}$$

Since $J_{\mu\nu}$ is approximately the identity matrix, we see that at equilibrium $h_\mu$ is close to either 0 or 1. Because our nonlinearity is a step function, its derivative vanishes at equilibrium, guaranteeing the stability of the fixed points (Appendix A.2). Consequently, when we solve Eq. (1), we expect to see $2^{N_h}$ stable fixed points in the regime $N_v \gg N_h$. This prediction is consistent with numerical simulations, as can be seen in Figure 1a.

## 2.2 Basins of Attraction

Although the fixed points are stable, that still leaves the question: how big are the basins of attraction? We'll assume that noisy input enters the network via the visible units, and initially all the $h_\mu$ are zero. How far from the fixed points can the input be and still be recalled perfectly?

Assuming the noise is additive, the initial values of the visible and hidden neurons are,

$$v_i(0) = \frac{1}{\sqrt{N_h}} \sum_{\mu=1}^{N_h} \xi_{i\mu} \Theta(h_{\mu,\text{target}} - \theta) + \epsilon_i^v \tag{12a}$$

$$h_\mu(0) = 0 \tag{12b}$$

where $h_{\mu,\text{target}} = 1$ if neuron $\mu$ encodes the target memory, and 0 otherwise (motivated by the fact that $h_\mu$ is close to either 0 or 1 at the fixed points; see Eq. (11)) and the noise, $\epsilon_i^v$ is independent and normally distributed with variance $\sigma_v^2$,

$$\epsilon_i^v \sim \mathcal{N}(0, \sigma_v^2). \tag{13}$$

To reach the target fixed point in both the hidden and visible unit space, the hidden neurons must evolve to their target values, $h_{\mu,\text{target}}$, before the visible units change much. That requires the visible units to evolve much more slowly than the hidden units, which we can guarantee by setting $\tau_h \ll \tau_v$ (see Appendix G ). With this condition, at a time $t$ satisfying $\tau_h \ll t \ll \tau_v$, $h_\mu(t)$ reaches equilibrium while $v_i(t)$ is still approximately equal to $v_i(0)$. Using Eq. (1a) with $dv_i/dt = 0$ along with Eq. (6), that equilibrium is given by

$$h_\mu(t) = \sum_\nu J_{\mu\nu} \Theta(h_{\nu,\text{target}} - \theta) + \frac{\sqrt{N_h}}{N_v} \sum_{i=1}^{N_v} \xi_{\mu i} \epsilon_i^v + \mathcal{O}(t/\tau_v). \tag{14}$$

Using Eq. (7a), we see that the first term is $\Theta(h_{\mu,\text{target}} - \theta) + \mathcal{O}(\sqrt{N_h/N_v})$. And the second term scales as $\sigma_v \sqrt{N_h/N_v}$. Thus, so long as

$$\sigma_v^2 \ll \frac{N_v}{N_h}, \tag{15}$$

$h_\mu(t)$ will be close to its target value when $t \ll \tau_v$. Since $v_i(t)$ is close to its target value at that time, it will stay close, and asymptotically the target pattern will be recovered. Given that $N_v \gg N_h$, the added noise can be very large without affecting recall. Thus, the basin of attraction is very large (see Figure 1b and Appendix D).

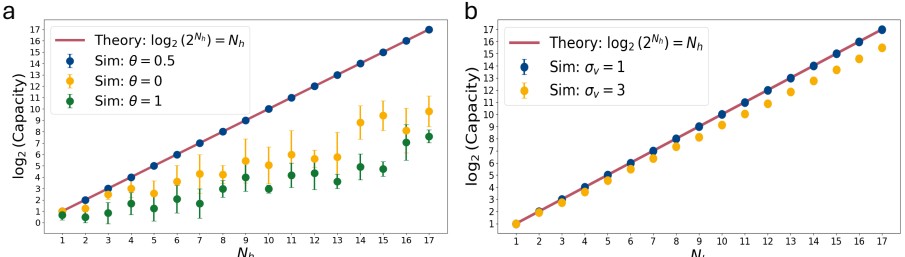

Figure 1: Capacity versus the number of hidden units, $N_h$, with $N_v = 100 N_h$ and $\tau_v = 20\tau_h$. (a) Capacity for different thresholds, $\theta$. The highest storage capacity is achieved when the threshold is set to its optimal theoretical value , $\theta = 0.5$. (b) The effect of noise in the visible layer ($\epsilon_i^v$ in Eq. (12a)), shown for different noise variances, demonstrates the large basin of attraction of the fixed points.

## 2.3 BIOLOGICAL PLAUSIBILITY

Compared to Krotov and Hopfield (2021), our model exhibits greater biological plausibility in several respects.

The activation function used here is local and keeps neuron activity within a biologically realistic range. In contrast, in Krotov and Hopfield (2021), Model A is not biologically plausible because the hidden neuron activities can grow to unrealistically large values as a consequence of the power-law activation, which does not reflect realistic neural firing. Models B and C, on the other hand, rely on non-local activation functions, softmax and spherical normalization, respectively, which are biologically implausible unless additional mechanisms are assumed.(see Appendix C for more details).

Although our theoretical analysis focuses on symmetric weights and a global threshold for all neurons, these assumptions are not restrictions of the model. Experimentally, we show that networks with asymmetric weights and heterogeneous thresholds also achieve stable recall. Allowing asymmetric

weights is important because exact symmetry is rarely observed in biological neural circuits. Similarly, heterogeneous thresholds capture the variability in neuron excitability across real neurons, and demonstrate that stable memory dynamics do not require finely tuned, uniform parameters. Together, these features indicate that our model better reflects realistic neural mechanisms while retaining associative memory functionality. Figure 9 in Appendix E shows representative recall examples for networks with asymmetric weights and heterogeneous thresholds that stored MNIST and CIFAR-10 images, respectively, using a learning rule similar to that discussed in Section 3.1.

# 3 RESULTS

## 3.1 LEARNING RULE

So far we have focused on storage capacity with fixed synaptic weights. A natural next step is to understand how these weights can be learned. In this section, we introduce a learning rule that reflects *compositional learning*: a small number of simple, reusable components can be combined to form complex patterns, and conversely, complex patterns can be decomposed into simpler components.

In steady-state, visible activity in Eq. (1) can be expressed as

$$\mathbf{v} = \frac{1}{\sqrt{N_h}} \sum_{\mu=1}^{N_h} \boldsymbol{\xi}_\mu s_\mu, \tag{16}$$

where $\boldsymbol{\xi}_\mu$ is the $\mu$-th column of $\xi \in \mathbb{R}^{N_v \times N_h}$, i.e., $(\boldsymbol{\xi}_\mu)_i = \xi_{i\mu}$. If only hidden neuron $\mu$ is active, the visible state equals $\boldsymbol{\xi}_\mu$. A visible memory is called *basic* if it corresponds to a single active hidden neuron, and *complex* if it is formed by the activation of multiple hidden neurons, i.e. a composition of several basic memories.

The goal of learning is to find a synaptic weight matrix $\xi$ and a threshold $\theta$ such that a set of target memories $\{\mathbf{v}_m \in \mathbb{R}^{N_v}\}_{m=1}^M$ approximately correspond to stable fixed points of the network dynamics, with $M \gg N_h$ (e.g., MNIST or CIFAR-10). This is achieved using the following optimization procedure,

$$(\xi, \theta) = \arg\min_{\xi, \theta} \sum_{m=1}^M \left\| \mathbf{v}_m - \frac{1}{\sqrt{N_h}} \sum_{\mu=1}^{N_h} \boldsymbol{\xi}_\mu \Theta\left( \frac{\sqrt{N_h}}{N_v} \boldsymbol{\xi}_\mu^\top \mathbf{v}_m - \theta \right) \right\|^2, \tag{17}$$

where we have replaced $s_\mu$ with its target steady-state value. This learning rule is identical to the one proposed in Radhakrishnan et al. (2020). We used Xavier initialization for the weights and approximated the threshold function $\Theta$ with a sharp sigmoid to allow gradient-based training.

## 3.2 EXPERIMENTS

Figure 2 shows recall results after storing 60,000 MNIST digits with $N_v = 784$ and $N_h = 50$. Despite the high correlation among patterns, the network learns 57913 unique minima corresponding to the 60,000 stored images. Variants of the same digit produce hidden representations that are distinct yet partially overlapping, and the recalled visible states remain recognizable.

Figure 3a shows the learned basic memories for the MNIST dataset, which correspond to the columns of $\xi$. As shown in Figure 3b, these basic memories are nearly orthogonal, consistent with Eq. (7a).

To evaluate the generality of the proposed learning rule beyond MNIST, we applied the same procedure to the CIFAR-10 dataset. In this case, $N_v = 3072 \, (3 \times 32 \times 32)$, and to compensate for the increased complexity of this dataset, we used a network with 500 hidden neurons, compared to 50 hidden neurons for MNIST. Figure 4 presents examples of the cues alongside their recalls. These results show that the network is able to reconstruct interpretable outputs from the learned representations, despite storing a large number of complex memories (50,000) and significantly violating the condition $N_v \gg N_h$. Importantly, these images are highly correlated, yet the network produces 49982 unique stable minima corresponding to the stored memories, with each memory representation being both stable and interpretable.

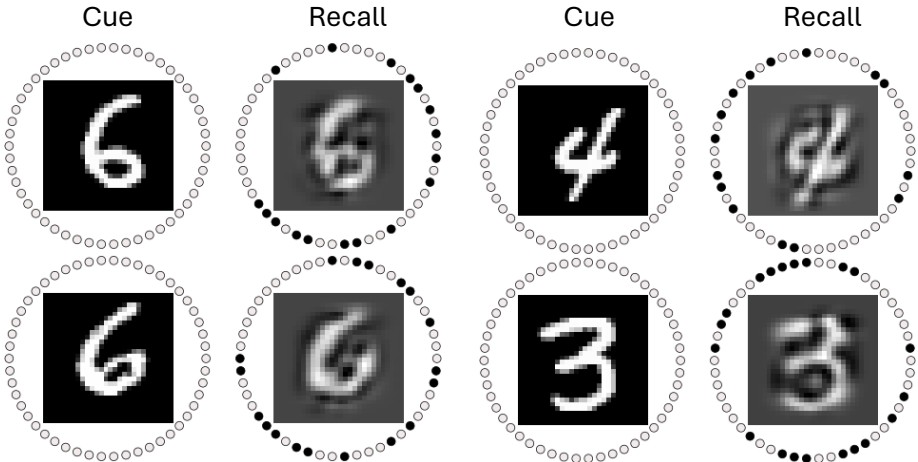

Figure 2: Examples of recall in a network with 50 hidden neurons that memorized 60,000 MNIST images. Hidden neurons are shown on the ring, and visible neurons are visualized as two-dimensional images. On the ring, black indicates high activity, and white indicates low activity. Highly correlated images of every digit, for instance, the digit 6 shown here, converge to unique but overlapping hidden representations.

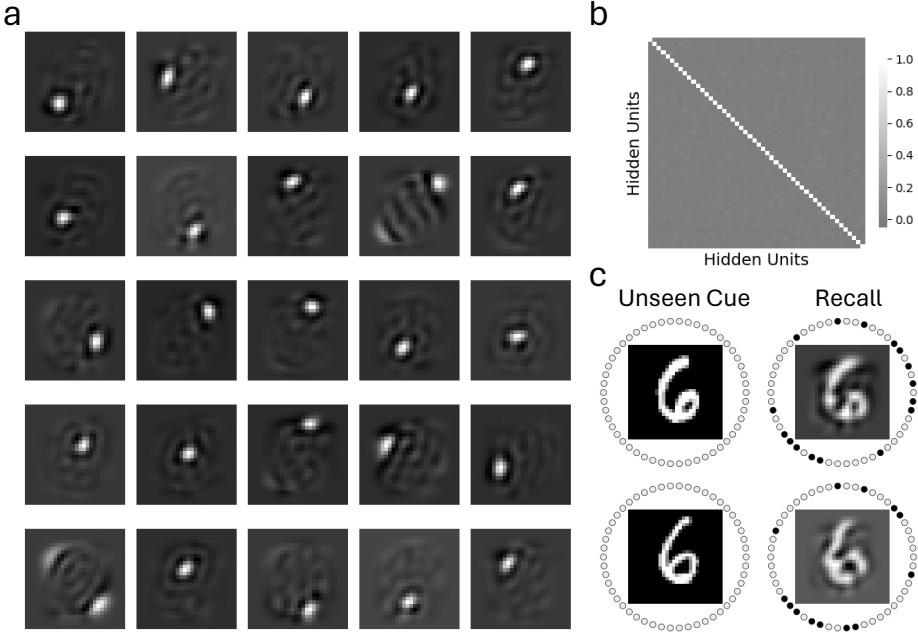

Figure 3: a) 25 (out of 50) columns of the learned weight matrix for MNIST images, which serve as basic memories, are shown as two-dimensional images. b) Correlation matrix of the basic patterns, which correspond to the hidden units. c) The network generalizes compositionally, associating unseen cues with interpretable fixed points.

The learned basic memories for CIFAR-10 images are shown in Figure 5a. They form a more heterogeneous set, yet remain nearly orthogonal, as shown in Figure 5b.

The network learns a global threshold of $\theta = 0.21$ for MNIST and $\theta = 0.43$ for CIFAR-10. Note that the statistics of the learned basic memories in MNIST and CIFAR-10 differ from one another

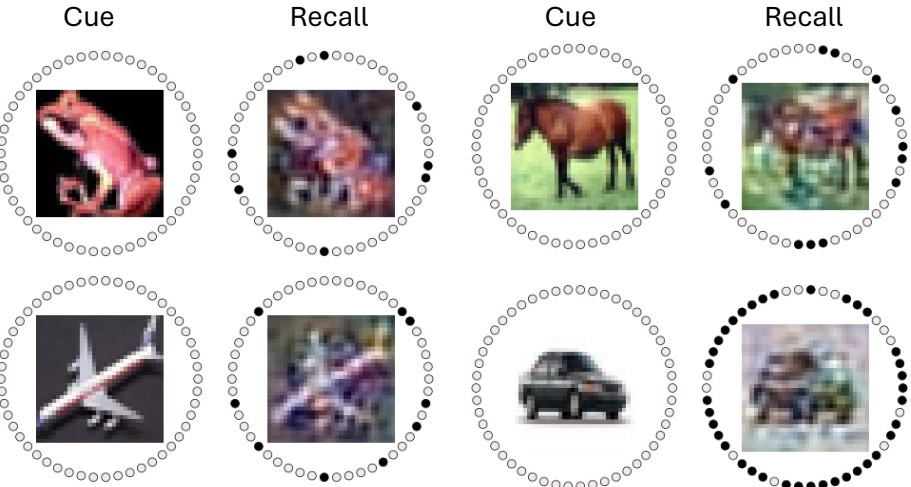

Figure 4: Examples of recall in a network with 500 hidden neurons that memorized 50,000 CIFAR-10 images. Hidden neurons are arranged on a ring (50 out of 500), while visible neurons are shown as two-dimensional images. On the ring, black indicates high activity, and white indicates low activity.

and from the normal distribution assumed in the theory, which explains the difference between the learned thresholds.

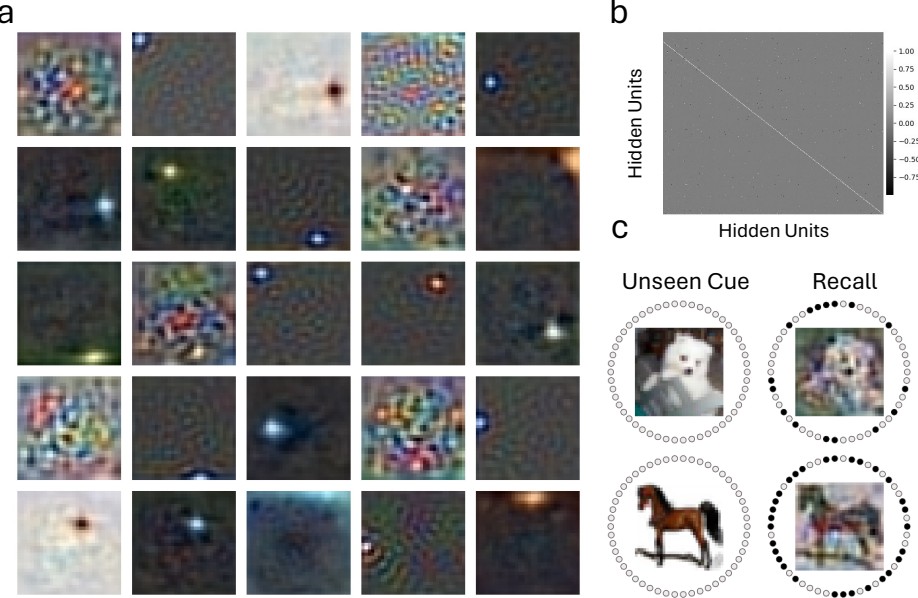

Figure 5: a) 25 (out of 500) columns of the learned weight matrix for CIFAR-10 images, which serve as basic memories, are shown as images. b) Correlation matrix of the basic patterns, which correspond to the hidden units. c) The network generalizes compositionally, associating unseen cues with interpretable fixed points.

For this system to function as an associative memory, new "unseen" cues should converge toward the approximately correct fixed points. For example, cues related to dogs should end up near fixed points associated with dogs, not horses. As shown in Figure 3c and Figure 5c, this behavior is indeed observed. The network converges to the nearest minimum of the energy landscape relative to the cue.

Importantly, when the learned basic memories are expressive enough, this nearest minimum can correspond to a stable representation that is very close to the unseen cue itself rather than to a memorized pattern. In other words, the network not only memorizes but also generalizes: the learned basic memories shape the energy landscape so that unseen inputs are mapped to stable attractors that capture their distinctive features.

For example, in Figure 3c, when two unseen images of the digit "6" are presented, the network converges to two distinct attractors that preserve the distinguishing details of each input while still sharing overlapping components in the hidden layer that identify them as class "6". This illustrates an advantage of our model: it supports both memorization and generalization through its learned basic components.

To quantitatively evaluate this behavior, we trained nonlinear classifiers on the recalled hidden and visible representations of the stored images, and tested them on the recalled representations of unseen images (Convolutional neural network (CNN) for visible representations and multilayer perceptron (MLP) for hidden representations). For comparison, we trained CNNs directly on the original stored images and tested them on the original unseen images as well. This allows us to assess the classifiability of the recalled visible representations by the associative memory network relative to the original memories. (see Appendix B for details of the classifiers)

For MNIST, classification accuracy is high for both the hidden and visible representations. This is a desirable property, as it indicates that the lower-dimensional hidden representations still preserve strong class discriminability. In MNIST, the raw pixel space itself carries strong class structure: two images of the same digit are highly correlated and closer to each other than images of different digits. Consequently, the hidden neurons retain this information almost perfectly, having structured and meaningful encodings in which correlated memories are represented close together and remain classifiable.

For CIFAR-10, classification accuracy is higher for the visible representations than for the hidden ones. This difference arises because, in CIFAR-10, two images of the same class (for example, dogs) are not necessarily correlated in raw pixel space, so the hidden layer, which is a nonlinear transformation of those pixels, does not exhibit a clear class structure. The classifier used for the visible representations in this analysis is a CNN, which, when trained on raw pixel data, first learns a nonlinear transformation that maps images of the same class close together in a learned feature space while separating images from different classes. After this transformation, classification is performed using a linear decision boundary in that space. This ability to internally build such class-specific representations explains why classification accuracy remains higher for the visible neurons.

Overall, the high performance on the visible representations for MNIST, and the reasonably high performance for CIFAR-10, demonstrates that the recalled representations remain class-discriminative, and that the associative memory preserves the essential structure of the data, enabling *compositional generalization* to unseen examples (for the CIFAR10 dataset, the classification accuracy can be increased by scaling up the associative memory, including $N_h$, the epoch size, the optimization step, and the number of training samples).

| Representation | MNIST Accuracy | CIFAR-10 Accuracy |
|---|---|---|
| Recalled Hidden Patterns | 95% | 40% |
| Recalled Visible Patterns | 98% | 56% |
| Original Images | 99% | 88% |

Table 1: Classification test accuracy of nonlinear classifiers trained and tested on recalled hidden and visible representations, as well as on the original images for comparison, for MNIST and CIFAR-10. Reasonably high accuracy on visible representations for both datasets demonstrates that the recalled representations remain highly class-discriminative, while the lower accuracy on hidden representations for CIFAR-10 reflects the lack of strong class structure in raw pixel space.

## 4 CONCLUSIONS

This work introduces a novel Dense Associative Memory Krotov and Hopfield (2021) that achieves exponential storage capacity in the number of hidden neurons, overcoming the limitations of previous

two-layer models. By using a threshold activation function, with a theoretically derived threshold, the network supports distributed hidden representations, allowing each hidden neuron to participate in multiple memories. This enables compositional storage of complex and correlated patterns, reducing redundancy while maintaining robust retrieval.

Notably, the network achieves exponential capacity, $2^{N_h}$, using only $N_h N_v$ parameters. In contrast, previous two-layer implementations were limited to a maximum capacity of $N_h$ (Krotov and Hopfield (2021)). As a result, in the proposed network, the number of memories per weight grows as $\frac{2^{N_h}}{N_h N_v} \approx 2^{N_h}$, while in previous implementations it is at best $\frac{1}{N_v}$. Even for complex datasets such as MNIST and CIFAR-10, networks with only 50 and 500 hidden units, respectively, were able to store tens of thousands of highly correlated memories and associate the vast majority of them with unique minima, whereas previous models could not store more memories than the number of hidden units (see AppendixC).

Beyond storing exponentially many memories, the network is also able to generalize to novel inputs. This behavior arises because the hidden layer encodes a set of basic memories that can be flexibly composed to represent previously unseen patterns, associating them with distinct minima rather than forcing retrieval toward the nearest stored pattern, while still producing meaningful, class-consistent representations. Our results are consistent with biological principles of feature learning, as embodied in hierarchical predictive coding models, which detect novelty and generalize by recombining features across successive levels of abstraction (Li et al. (2025); Salvatori et al. (2021)). This mechanism further illustrates that learning the underlying compositional structure of naturalistic data enables a biological associative memory to effectively support both memorization and generalization.

The model is biologically grounded, relying solely on standard pairwise synapses and a local activation function. We also provide evidence that it achieves stable recall even in the presence of asymmetric weights and heterogeneous neuronal thresholds. Moreover, the hidden layer forms low-dimensional representations that preserve class-discriminative information, placing memories with shared components close together in activity space. This structured organization supports efficient nonlinear decoding.

Overall, this work establishes a new regime for associative memory that combines high capacity, robust recall, compositional and interpretable representations, and biological plausibility. It provides a theoretical foundation for scalable memory systems that bridge neuroscience models with modern machine learning architectures.

Future work will focus on developing a biologically plausible learning rule and on examining the model's capacity under additional biological constraints, including sparse connectivity and adherence to Dale's law.

## ACKNOWLEDGMENTS

We thank Brendan A. Bicknell and Reidar Riveland for helpful discussions. We acknowledge the following funding sources: Gatsby Charitable Foundation (GAT3850) to MSK and PEL. The results presented here were obtained while Dmitry Krotov was employed by IBM Research. At the time of the camera-ready submission Dmitry Krotov is no longer employed by IBM Research.

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

# A  Technical Appendices and Supplementary Material

## A.1  Distributional Properties of $\zeta_{\mu\nu}$

Combining Eqs. (6) and (7a), we see that $\zeta_{\mu\nu}$ is given by

$$\zeta_{\mu\nu} = \frac{1}{\sqrt{N_v}} \sum_{i=1}^{N_v} \xi_{\mu i}\xi_{i\nu} - \sqrt{N_v}\delta_{\mu\nu}\,. \tag{18}$$

where, recall, the $\xi_{\mu i}$ are independent, zero mean unit variance Gaussian random variables (see Eq. (3)). Thus, for $\mu \neq \nu$, each product $\xi_{\mu i}\xi_{i\nu}$ is a zero-mean, unit variance random variable. By the central limit theorem, the sum of these $N_v$ independent terms converges in distribution to a Gaussian, so for $\mu \neq \nu$ we have

$$\zeta_{\mu\nu} \xrightarrow{d} \mathcal{N}(0,1)\,. \tag{19}$$

For $\mu = \nu$, we write

$$\zeta_{\mu\mu} = \frac{1}{\sqrt{N_v}} \sum_{i=1}^{N_v} (\xi_{\mu i}^2 - 1)\,. \tag{20}$$

By the central limit theorem, this too is Gaussian, but with a slightly larger variance,

$$\zeta_{\mu\mu} \xrightarrow{d} \mathcal{N}(0, \mathrm{Var}[\xi^2]) = \mathcal{N}(0,2)\,. \tag{21}$$

Combining Eqs. (19) and (21), we arrive at

$$\zeta_{\mu\nu} \xrightarrow{d} \mathcal{N}(0, 1 + \delta_{\mu\nu})\,. \tag{22}$$

Now consider the random variable $q_\mu$ defined as

$$q_\mu = \frac{1}{\sqrt{N_v}} \sum_{\nu=1}^{N_h} \zeta_{\mu\nu}s_\nu, \tag{23}$$

This is a random variable with respect to the index $\mu$ with $s_\nu$ fixed. Its variance is given by

$$\mathrm{Var}\left[\frac{1}{\sqrt{N_v}} \sum_{\nu=1}^{N_h} \zeta_{\mu\nu}s_\nu\right] = \frac{1}{N_v} \sum_{\nu=1}^{N_h} s_\nu^2 \mathrm{Var}[\zeta_{\mu\nu}] = \frac{1}{N_v} \sum_{\nu=1}^{N_h} s_\nu^2(1 + \delta_{\mu\nu}) \tag{24}$$

where we used the fact that the $\zeta_{\mu\nu}$ are independent random variables with mean 0 and variance given in Eq. (22).

## A.2  Stability of the fixed points

The stability of fixed points is determined by the Jacobian of the system. Grouping the variables into $(\mathbf{v}, \mathbf{h})$, corresponding to the visible and hidden units respectively, the Jacobian, denoted $\mathbf{A}$, has the block structure

$$\mathbf{A} = \begin{bmatrix} \mathbf{A}_{vv} & \mathbf{A}_{vh} \\ \mathbf{A}_{hv} & \mathbf{A}_{hh}\ . \end{bmatrix} \tag{25}$$

For the diagonal blocks, consider first the visible units. We have

$$\frac{\partial \dot{v}_i}{\partial v_j} = \begin{cases} -1, & j = i, \\ 0, & j \neq i, \end{cases} \quad \Rightarrow \quad \mathbf{A}_{vv} = -\mathbf{I}_{N_v}, \tag{26}$$

where $\mathbf{I}_{N_v}$ is the $N_v \times N_v$ identity matrix. Similarly, for the hidden units,

$$\frac{\partial \dot{h}_\mu}{\partial h_\nu} = \begin{cases} -1, & \nu = \mu, \\ 0, & \nu \neq \mu, \end{cases} \quad \Rightarrow \quad \mathbf{A}_{hh} = -\mathbf{I}_{N_h}, \tag{27}$$

where $\mathbf{I}_{N_h}$ is the $N_h \times N_h$ identity matrix.

For the off-diagonal blocks, the derivative of the Heaviside step function in Eq. (2) is zero almost everywhere,

$$\Theta'(z) = 0, \qquad z \neq 0. \tag{28}$$

Therefore, away from threshold crossings ($h_\mu \neq \theta$) in Eq. (1a),

$$\frac{\partial \dot{v}_i}{\partial h_\mu} = 0 \quad \Rightarrow \quad \mathbf{A}_{vh} = \mathbf{0}. \tag{29}$$

The hidden dynamics depend explicitly on the visible variables:

$$\frac{\partial \dot{h}_\mu}{\partial v_i} = \xi_{\mu i}, \quad \Rightarrow \quad \mathbf{A}_{hv} = (\xi_{\mu i}). \tag{30}$$

Putting everything together, the Jacobian is lower-triangular,

$$\mathbf{A} = \begin{bmatrix} -\mathbf{I}_{N_v} & \mathbf{0} \\ \mathbf{A}_{hv} & -\mathbf{I}_{N_h} \end{bmatrix}. \tag{31}$$

The eigenvalues of a triangular matrix are its diagonal entries, which in this case are all equal to $-1$. Hence, all fixed points of the dynamics are stable.

## B  DETAILS OF THE CLASSIFIERS

Table 2: Architecture and Parameters of the CNN Classifier

| Block | Layer Type | Channels / Filters | Kernel / Pool | Activation |
|---|---|---|---|---|
| Conv Block 1 | $2 \times$ Conv2D + BatchNorm2D | $3 \to 32$ | $3 \times 3$, MaxPool(2) | ReLU |
| Conv Block 2 | $2 \times$ Conv2D + BatchNorm2D | $32 \to 64$ | $3 \times 3$, MaxPool(2) | ReLU |
| Conv Block 3 | $2 \times$ Conv2D + BatchNorm2D | $64 \to 128$ | $3 \times 3$, MaxPool(2) | ReLU |
| Flatten | – | Auto-computed ($f$) | – | – |
| Fully Connected 1 | Linear | $f \to 128$ | – | ReLU |
| Fully Connected 2 | Linear | $128 \to N_{classes}$ | – | Softmax |

Table 3: Architecture and Parameters of the MLP Classifier

| Layer | Type | Dimensions / Units | Activation |
|---|---|---|---|
| Input | Linear | Input dimension $= d$ | – |
| Hidden Layer 1 | Linear | $d \to 256$ | ReLU |
| Hidden Layer 2 | Linear | $256 \to 128$ | ReLU |
| Output Layer | Linear | $128 \to N_{classes}$ | Softmax |

## C  COMPARISON WITH PREVIOUS DENSE ASSOCIATIVE MEMORY NETWORKS

Krotov and Hopfield (2021) proposed a two-layer associative memory defined as

$$\tau_v \frac{dv_i}{dt} = -v_i + \sum_{\mu=1}^{N_h} w_{i\mu} f(h_\mu), \tag{32a}$$

$$\tau_h \frac{dh_\mu}{dt} = -h_\mu + \sum_{i=1}^{N_v} w_{\mu i} g(v_i), \tag{32b}$$

where $w_{\mu i} = w_{i\mu}$ and each $\mathbf{w}_\mu \in \mathbb{R}^{N_v}$ represents a stored memory. Three choices for the nonlinearities $f$ and $g$ were introduced in Krotov and Hopfield (2021), summarized in Table 4.

We evaluate their recall performance (Models A, B, and C) together with our proposed model. Model A is tested using $f(h_\mu) = h_\mu^5$ (A (i)) and $f(h_\mu) = h_\mu^{10}$ (A (ii)).

Despite differences, all three models operate under the same effective mechanism: during recall, a single hidden neuron becomes strongly active while the others remain suppressed, allowing only one memory per hidden neuron.

In contrast, our nonlinearity produces a fundamentally different recall regime. Multiple hidden neurons remain active simultaneously, enabling each hidden neuron to encode multiple stored patterns. This results in a dramatic increase in storage capacity: with only fifty hidden neurons, our model successfully stores all MNIST images (60,000) with high recall accuracy. By comparison, Models A, B, and C are limited to 50 memories and often fail to recall reliably (e.g., Model A (i) and Model C), as shown in Table 4.

And from a biological perspective, the nonlinearity used in Model A is not plausible, because the power-law activation causes hidden neuron activity to reach unrealistically high values during recall. Models B and C also rely on non-local activation functions, which would require additional circuit mechanisms to implement. In contrast, our model maintains bounded activity, and the nonlinearity is fully local.

| Model | $f(h_\mu)$ | $g(v_i)$ | $N_h$ | # Stored Memories | Recall Performance |
|---|---|---|---|---|---|
| A (i) | $h_\mu^5$ | $\mathrm{sign}(v_i)$ | 50 | 50 | 12% |
| A (ii) | $h_\mu^{10}$ | $\mathrm{sign}(v_i)$ | 50 | 50 | 84% |
| B | $\dfrac{e^{h_\mu}}{\sum_\nu e^{h_\nu}}$ | $v_i$ | 50 | 50 | 90% |
| C | $h_\mu^5$ | $\dfrac{v_i}{\sqrt{\sum_j v_j^2}}$ | 50 | 50 | 2% |
| **Our model** | $\Theta(h_\mu - \theta)$ | $v_i$ | 50 | **60,000** | **98%** |

Table 4: Nonlinearities used in the Dense Associative Memory models from Krotov and Hopfield (2021) and in our model, and a comparison of their recall performance. Recall performance is the percentage of recalled digits that are classified correctly.

## D  NOISE-INDUCED BIT FLIPS

As pointed out in Secs. 2.1 and 2.2, if the noise – associated either with finite $N_v$ corrections to $J_{\mu\nu}$, with perturbations to the initial value of the visible units, or both – is too large, the target solution can be destabilized via "bit flips" in the hidden units. Here we quantify this, and in particular show that the probability of a bit flip falls off rapidly as the number of visible units, $N_v$, increases.

From Eqs. (9) and (14), with $\theta$ set to 0.5, we have

$$s_\mu(t) = \Theta\left( s_{\mu,\mathrm{target}} + q_{\mu,\mathrm{target}} + \frac{\sqrt{N_h}}{N_v} \sum_{i=1}^{N_v} \xi_{\mu i} \epsilon_i^v + \mathcal{O}(t/\tau_v) - 0.5 \right), \tag{33}$$

where $q_{\mu,\mathrm{target}}$ is a normally distributed random variable with variance given in Eq. (24), but with $s_\nu$ replaced by $s_{\nu,\mathrm{target}}$. As discussed in the paragraph following Eq. (14), the third term in parentheses has variance $\sigma_v^2 N_h/N_v$. Thus, in the limit $\tau_v \gg \tau_h \gg t$, we can approximate

$$s_\mu(t) = \Theta\left( s_{\mu,\mathrm{target}} + z_{\mu,\mathrm{target}} - 0.5 \right), \tag{34}$$

where $z_{\mu,\mathrm{target}}$ is normally distributed with variance given by, at most,

$$\mathrm{Var}[z_{\mu,\mathrm{target}}] = \frac{N_{a,\mathrm{target}} + 1 + \sigma_v^2 N_h}{N_v} . \tag{35}$$

Here $N_{a,\text{target}}$ is the number of active target hidden neurons, and the "at most" qualifier is because we should include the "+1" term only if $s_{\mu,\text{target}} = 1$ (see Eq. (24)).

The probability of a single bit flip depends in a relatively simple way on the variance of $z_{\mu,\text{target}}$. However, the dependence on the number of active target neurons and the "+1" makes it nontrivial to determine the probability that there are no bit flips at all. We therefore bound that probability by focusing on the upper bound on the variance of $z_{\mu,\text{target}}$, which is given by

$$\sigma_z^2 \equiv \frac{N_h + 1 + \sigma_v^2 N_h}{N_v} . \tag{36}$$

With this definition, the probably of a single bit flip , which is the probability that $z_{\mu,\text{target}}$ is either greater than 1/2 or less than -1/2, depending on whether $s_{\mu,\text{target}}$ is 0 or 1, is bounded by

$$P_{1\text{ bit flip}} \leq 1 - \Phi(1/2\sigma_z) \tag{37}$$

where $\Phi$ is the cumulative normal function. The probability of no bit flips is, then, bounded by

$$P_{\text{no bit flips}} = (1 - P_{1\text{ bit flip}})^{N_h} \geq \Phi(1/2\sigma_z)^{N_h} . \tag{38}$$

To write this in a more interpretable form, note that

$$1 - \Phi(a) = \int_a^\infty dx\, \frac{e^{-\frac{x^2}{2}}}{\sqrt{2\pi}} = \int_0^\infty dy\, \frac{e^{-\frac{a^2}{2} - ay - \frac{y^2}{2}}}{\sqrt{2\pi}} \leq \int_0^\infty dy\, \frac{e^{-\frac{a^2}{2} - ay}}{\sqrt{2\pi}} = \frac{e^{-\frac{a^2}{2}}}{a\sqrt{2\pi}} . \tag{39}$$

Inserting this into Eq. (40) then gives us the bound

$$P_{\text{no bit flips}} \geq \left(1 - 2\sigma_z \frac{e^{-1/8\sigma_z^2}}{\sqrt{2\pi}}\right)^{N_h} \geq 1 - N_h \sigma_z \frac{e^{-1/8\sigma_z^2}}{\sqrt{\pi/2}} \tag{40}$$

where the second inequality follows from Jensen's inequality.

This probability rapidly approaches 1 as $\sigma_z$ becomes small. In this regime, both the stability of the fixed point and correct recall are ensured.

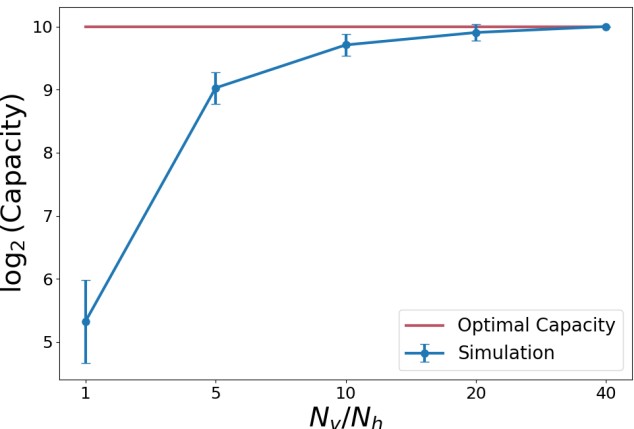

Figure 6: Capacity versus the ratio between visible and hidden neurons, for a fixed value of $N_h = 10$.

To determine how many hidden neurons are required for real world memories with diverse statistics, note that memories from an $N_v$-dimensional space are recalled within an at most $N_h$-dimensional subspace spanned by the $N_h$ basic memories defined by the hidden to visible weights. If this subspace is not expressive enough, the reconstructed images will not be recognizable, particularly for complex datasets such as CIFAR-10. Consequently, the number of hidden neurons must be sufficiently smaller than the number of visible neurons to guarantee stable recall, but it must also be

large enough to represent the statistical structure of the stored memories so that the reconstructions remain recognizable.

Figure 7 and Figure 8 show representative recall examples on MNIST and CIFAR-10, for networks with an insufficient number of hidden neurons ($N_h = 16$).

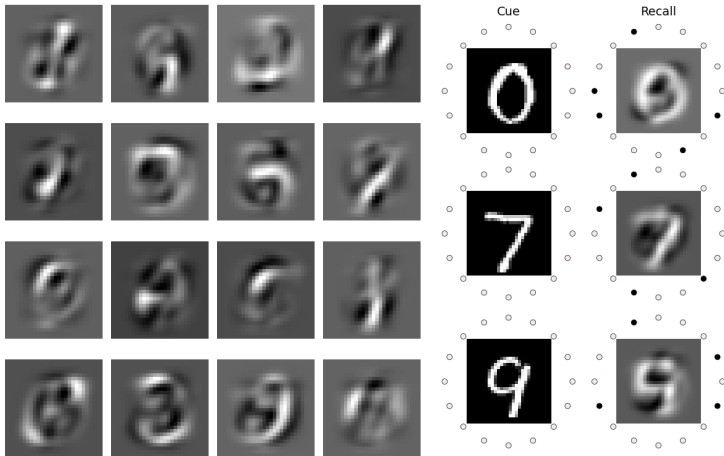

Figure 7: Learned basic memories (columns of learned weight matrix) and representative recall examples for MNIST, for a network with an insufficient number of hidden neurons ($N_h = 16$).

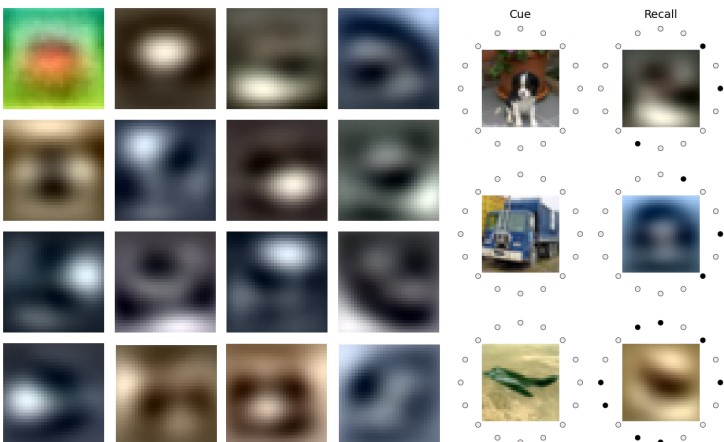

Figure 8: Learned basic memories (columns of learned weight matrix) and representative recall examples for CIFAR-10, for a network with an insufficient number of hidden neurons ($N_h = 16$).

# E    ASYMMETRIC WEIGHTS AND HETEROGENEOUS THRESHOLDS

Figure 9 shows representative recall examples for networks with asymmetric weights and heterogeneous neuron thresholds on both MNIST and CIFAR-10.

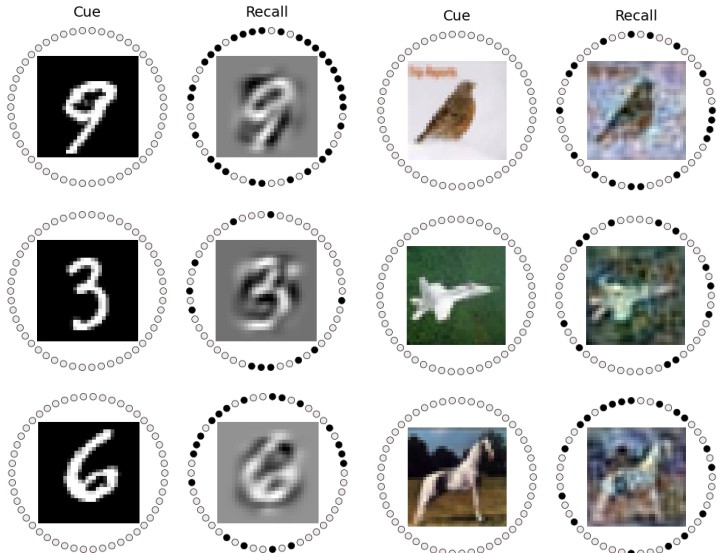

Figure 9: Representative recall examples for networks with asymmetric weights and heterogeneous neuron thresholds. On the left are examples of cue and recall for a network that stored 60,000 MNIST images with 50 hidden neurons, and on the right are examples for a network that stored 50,000 CIFAR-10 images with 500 hidden neurons.

## F    SIGMOID ACTIVATION FUNCTIONS

In both the theory and the experiments, we used the Heaviside step function for the activation of hidden neurons. A smooth step function was used only during optimization.

However, the Heaviside function is not the only valid choice of activation. In fact, any sigmoid function that has three intersections with the identity line as showin in Figure 10a where the middle intersection at $1/2$ is an unstable fixed point of (8), and the intersections near $0$ and $1$ are stable fixed points, is a valid activation function that guarantees stable recall as shown in .

We demonstrated this experimentally by showing that recall remains perfect even when a smooth sigmoid is used in place of the Heaviside function, as shown in Figure 10b.

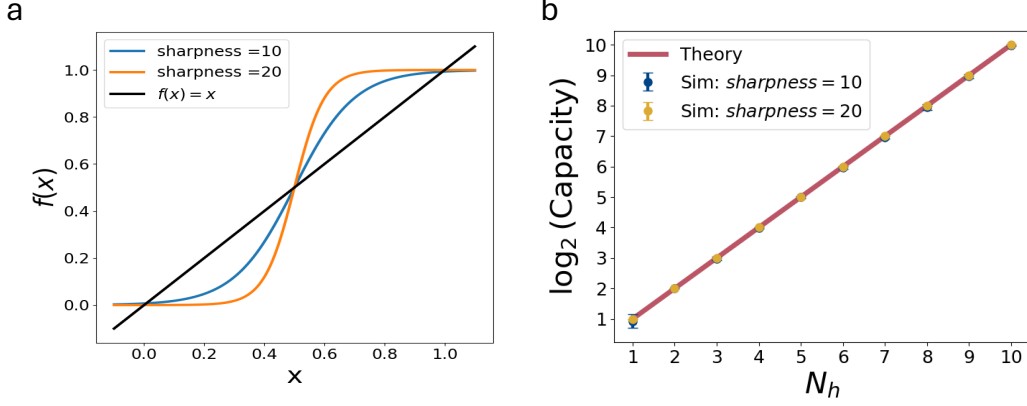

Figure 10: a) Sigmoid nonlinearities with different sharpness, each having three intersections with the identity line. b) Both nonlinearities produce stable recall, and the capacity remains exponential.

## G   THE TIME CONSTANT OF HIDDEN AND VISIBLE NEURONS

Figure 11 shows the impact of $\tau_v/\tau_h$, the ratio of the time constant between visible and hidden neurons on recall performance. As discussed in Section 3.2 on the basin of attraction, the visible neurons must be sufficiently slower than the hidden neurons because, during the cue, only the visible neurons receive input while the hidden neurons are initialized to zero. The visible neurons therefore need to evolve slowly enough to allow the hidden neurons to reach their correct steady state before the visible pattern changes significantly. Somewhat surprisingly, a factor of only 4 is enough to ensure perfect recall.

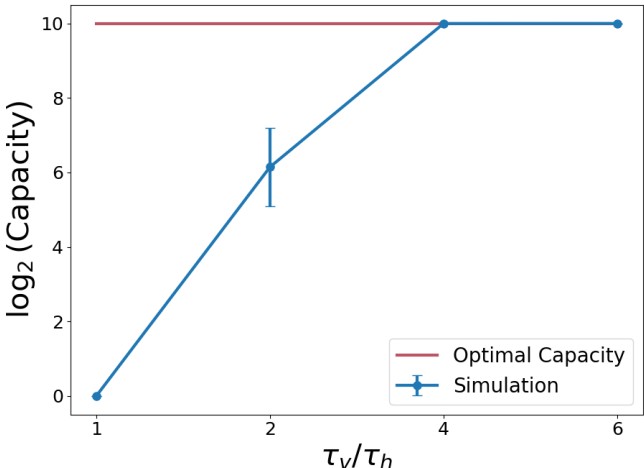

Figure 11: Capacity versus the ratio between the time constant of visible and hidden neurons for $N_h = 10$ and $N_v = 100N_h$.

