# OpenReview forum: "A Biologically Plausible Dense Associative Memory with Exponential Capacity"
_ICLR.cc/2026/Conference — ICLR 2026 Poster_

### Official Review · Reviewer_1mum · 2025-10-26

**Soundness:** 2
**Presentation:** 2
**Contribution:** 2
**Rating:** 2
**Confidence:** 4

**Summary:**

This paper presents a 2-layer associative memory network that extends Krotov and Hopfield (2021). The key innovation is a replacement of the softmax/WTA hidden layer nonlinearity with a Heaviside step function, which effectively enables multiple hidden neurons to represent a memory instead of forming grandmother cells. The paper performed theoretical study of the capacity of the network and found that the network can achieve $2^{N_h}$ capacity with certain assumptions (e.g., $N_v >> N_h$), and it also studied the size of the attractor basin.

**Strengths:**

- It proposes a simple yet effective fix to a known limit of Krotov and Hopfield 2021; theoretical proof and experiment analysis demonstrates the effectiveness of this proposal
- The theoretical analysis is correct, thorough and clearly presented

**Weaknesses:**

Conceptually, I feel this paper has the following weaknesses:
- The idea, though effective, is incremental to Krotov and Hopfield 2021. This is exacerbated by the fact that Krotov and Hopfield 2021 already discussed different possibilities of hidden/sensory nonlinearities so this paper does not serve as a generalization.
- Despite the point about, there is still opportunity for this idea to be influential if the authors could extend it along the lines of biological plausibility (e.g., what if the weights are not symmetric?) and ML applications (e.g., how can a transformer benefit from the step function nonlinearity if it can be plugged into attention?). In other words, the paper can go 'wide' as it is incremental from a 'depth' point of view
- The claimed biological plausibility is a direct inheritance of the biological plausible implmentation of Krotov and Hopfield 2021; no novel plausible implementation is introduced. The theoretically optimal $\theta=0.5$ may even be too strict as a biological constraint. Therefore I suggest the authors not to sell this point too much (for example in the title).

In addition, I also feel this paper lacks analytical and experimental depth:
- It lacks a direct comparison to other nonlinearities (e.g., models in Krotov and Hopfield 2021) - I understand theoretically it should have a bigger capacity, but a complete result section needs explicit experimental results to support the theory
- Many experimental setting violates assumptions in theory. For example, in experiments, the $\Theta$ function is a smoothed step-function; in both MNIST and CIFAR experiments $N_v$ is not much bigger than $N_h$.
- The capacity results do not seem consistent with the theory: for example, 55k unique minima is far less than $2^{N_h}=2^50$. This makes me wonder the importance of the violated theoretical assumptions above.

**Questions:**

- After training on MNIST/CIFAR10, how many distinct stable attractors exist beyond the training set, and how close is that to the theoretical $2^{N_h}$? Can't you experiment with a smaller $N_h$ e.g., 16, so the theoretical capacity is closer to the training set size?
- How sensitive is stability / capacity to smoothing the Heaviside (which you actually use in training) and to adding noise / heterogeneity in thresholds?
- How crucial is $\tau_h << \tau_v$? Can you show recall robustness curves as a function of this ratio?
- Can you concretely map your hidden code / retrieval dynamics onto a transformer attention head or a feedforward block, and/or show a toy transformer experiment?
- Can you tone down / clarify the “biological plausibility” claim, or alternatively provide evidence that asymmetric weights or mild threshold heterogeneity don’t affect capacity?

---

> ### Author Response · Authors · 2025-11-21
> **Authors Response to the Reviewer 1mum - Part 1**
>
> We thank the reviewer for their valuable time and for providing instructive feedback that has helped us improve our work. In the following, we address all questions and concerns, and we have updated the manuscript accordingly. The revised version is now ready for review.
>
> We would be grateful if the reviewer could consider increasing their score should they find the updated work satisfactory.
>
> ### W1. Regarding the novelty of this work:
> While different nonlinearities are discussed in that earlier work, as stated in Eq.9 of that paper, all of those nonlinearities lead to a capacity bounded by $N_h$. In contrast, we introduce a fundamentally different nonlinearity that yields an exponential capacity $2^{N_h}$.
>
> To further elaborate on this point, we added Appendix C, which provides a direct comparison. We show that with $N_h = 50$, using all the nonlinearities introduced in Krotov and Hopfield 2021, Models A, B, and C, one cannot store more than 50 memories, and even for that small number the models often fail to recall reliably (for example Model A i and Model C). However, with the same number of hidden neurons, our model stores the entire MNIST dataset of 60{,}000 images, and the recall remains high.
>
> Thus, the importance of the new nonlinearity lies in enabling memories to be stored compositionally, which represents a fundamentally different mechanism compared to earlier work.
>
>
> ### W2-a. Regarding asymmetric weights:
> This is very constructive feedback. A network with asymmetric weights can also achieve stable recall, demonstrated experimentally for both the MNIST and CIFAR-10 datasets. We added Appendix E to include these results.
>
> ### W2-b. Regarding extending this work to attention mechanisms in transformers:
> We are grateful to the reviewer for this insightful suggestion. Although this is currently beyond the focus and scope of the present work, it is indeed a very interesting direction for future research, and we definitely plan to consider it.
>
>
> ### W3. Regarding biological plausibility:
>
> This is an important point. In summary, the network is more biologically plausible than Krotov and Hopfield 2021 for the followig reasons:
>
> 1. The activation function used here is local and keeps neuron activity within a biologically realistic range. In contrast, in Krotov and Hopfield (2021), Model A is not biologically plausible because the hidden neuron activities can grow to unrealistically large values as a consequence of the power-law activation, which does not reflect realistic neural firing. Models B and C, on the other hand, rely on non-local activation functions, spherical normalization and softmax, respectively, which are biologically implausible unless additional mechanisms are assumed.
>
>
> 2. Networks with asymmetric weights can also achieve stable recall, showing that symmetry constraints are not strictly necessary.
>
> 3. Networks with heterogeneous thresholds across neurons can also maintain stable fixed points, indicating that a shared threshold is not required.
>
> For points 2 and 3, we demonstrate experimentally on both the MNIST and CIFAR-10 datasets that memories can be successfully stored and recalled when learning both weights and thresholds without imposing symmetry or a common threshold. We added Appendix E to include these results.
>
>
> ### W4. Regarding comparison to Krotov and Hopfield (2021):
> In response to this feedback, we added Appendix C. Additionally, we provided a clarification on the fundamental difference between the current nonlinearity and the previous ones in response to W1 and W3.
>
> ### W5-Part a. Regarding the smoothed step function:
> This is an important point to clarify. In both the theory and the experiments, we used the Heaviside step function for the activation of hidden neurons. Only when we minimized the loss in Eq.17, we replaced the Heaviside step function with a (slightly) smoother function.
>
> However, the Heaviside step function is not the only valid choice for activation. Indeed, we could have used any activation function $f(x)$ so long as it had the following properties: 1) $f(x)$ is continuous; 2) $f(x)$ is monotonic non-decreasing; 3) the fixed points of the equation $x=f(x)$ occur at $x=\{0, x_1, 1\}$ with $0 < x_1 < 1$; 4) $df(x_1)/dx_1 > 0$.
>
> We added Appendix F and demonstrated experimentally that recall remains perfect even when using a smooth function instead of the Heaviside function.

---

> > ### Author Response · Authors · 2025-11-21
> > **Authors Response to the Reviewer 1mum - Part 2**
> >
> > ### W5- Part b. Regarding the ratio $\frac{N_v}{N_h}$:
> >
> >
> > This is an important observation. We added Appendix D clarifying that the probability of correct recall is
> > $$
> > P_\text{no flip} = \Phi(1/2\sigma_z)^{N_h}$$
> > where $\Phi$ is the cumulative distribution function of the standard normal distribution. This probability rapidly approaches 1 as the ratio $N_h/N_v$, and consequently $\sigma_z$, becomes smaller (please see Eq.34 in Appendix D). As we show in Appendix D, even when $N_v = 10 N_h$, exponentially many fixed points remain stable.
> >
> > To determine the number of hidden neurons required for real-world memories with diverse statistics, note that memories from an $N_v$-dimensional space are recalled within at most an $N_h$-dimensional subspace spanned by the $N_h$ basis memories defined by the hidden-to-visible weights. If this subspace is not sufficiently expressive, the reconstructed images will not be recognizable, particularly for complex datasets such as CIFAR10.
> >
> > In summary, the number of hidden neurons must be small enough relative to the number of visible neurons to guarantee stable recall, yet large enough to capture the statistical structure of the stored memories so that reconstructions remain recognizable.
> >
> > Finally, to address the ratio $N_h / N_v$ for CIFAR10, we updated our experiments to use the full RGB images of dimension $3072$, while keeping $N_h = 500$, reducing the ratio to approximately $1/6$. Appendix D shows examples of recalled representations for CIFAR10 and MNIST when the number of hidden neurons is insufficient.
> >
> >
> >
> >
> > ### W6. Regarding the $2^{N_h}$ capacity:
> > We are very thankful to the reviewer for raising this important point of clarification. The number of unique minima reported in the results corresponds to the number of unique minima associated with the stored memories, not to all existing minima in the network. We have revised the manuscript to clarify this distinction.
> >
> > In reality, there are exponentially many more stable minima in the network. Once a set of complex memories is stored, the network learns a set of basic memories. Each pattern of activity that can be composed from these basic memories represents a stable minimum, even if the network has never explicitly seen that minimum during training. We have also added new experiments to further demonstrate the existence and usefulness of these additional minima. Figure 3c, Figure 5c , and Table 1 now include a detailed discussion of these results.
> >
> > ### Q1.
> > Please see our response to W5-b and W6.
> >
> > ### Q2.
> > Please see our response to W5-a.
> >
> >
> > ### Q3. Regarding the effect of $\frac{\tau_v}{\tau_h}$ ratio:
> >
> > This is an important point. We have added Appendix G with results showing that having only $\frac{\tau_v}{\tau_h} = 4$, i.e., the visible neurons being about four times slower than the hidden neurons, already allows for stable recall of almost all memories, although a high recall percentage begins to appear even with $\frac{\tau_v}{\tau_h} = 2$.
> >
> >
> > ### Q4.
> > Please see our response to W2-b.
> >
> > ### Q5.
> > Please see our response to W2-a, W3 and W5-a. In particular, we're going with option B: we provide evidence that asymmetric weights or mild threshold heterogeneity don’t affect capacity.
> >
> > Once again, we thank the reviewer for their time and valuable feedback, which greatly helped us improve the clarity and presentation of our work. We hope that our responses have fully addressed all of their questions and concerns.

---

> > > ### Comment · Reviewer_1mum · 2025-11-22
> > >
> > > Thank you authors for addressing my questions and updating the paper. I'm generally happy to raise the score. However, there are few minor points, mostly about presentation of the paper, that I suggest the authors to consider:
> > >
> > > - Your 3 bullet points on W3. Regarding biological plausibility looks very clear to me, and I feel they are important to support the main argument on a more plausible dense associative memory, which you quite heavily emphasized in your title and intro. I suggest you include a dedicated subsection under you Model section to include these discussions. They will help establish your model's better plausibility than Krotov and Hopfield 2021, so that other readers will not wonder "what is novel about the biological plausibility?" just like I did. You may also consider adding a small diagram; but that's up to you.
> > > - The structure of the paper could be further improved. You now include everything under Results. I suggest you put everything up to Learning Rule under Model/Method, and then put MNIST/CIFAR experiments under a Results section. This is more standard and makes the paper easier to follow
> > > - I'm surprised by results in figure 3c and 5c - indeed they converge to dogs and horses, but what's surprising to me is that the recall didn't converge to a dog in the training set, or an average dog (same for 6), but to a blurry version of itself, which I suppose should not be the learned fixed points because it is unseen. Why is this? I don't think this will happen for any single-layer associative memory models (i.e., if you give them an unseen dog/6, they will converge to a seen or average one), so I'm wondering if this is due to the latent representation from the 2-layer structure. Maybe something worth discussing.

---

> > > > ### Author Response · Authors · 2025-11-23
> > > > **Authors Response to the Reviewer 1mum - Part 3**
> > > >
> > > > We gratefully acknowledge your thoughtful feedback and the time you invested in re-examining our manuscript.
> > > >
> > > >  We incorporated all your suggestions, and the revised version is now ready for your review.
> > > >
> > > > ### Regarding the subsection on biological plausibility:
> > > > We agree with your suggestion and added a dedicated subsection under the Model section on BIOLOGICAL PLAUSIBILITY.
> > > >
> > > > ### Regarding the paper structure:
> > > > We have revised the structure according to your constructive suggestion. The MODEL and RESULTS are now separate sections. Under RESULTS, we now have LEARNING RULE and EXPERIMENTS subsections.
> > > >
> > > >
> > > > ### Regarding the unseen images:
> > > > Your observation is correct. The network converges to the nearest minimum of the energy landscape relative to the cue. When the learned basic memories are sufficiently expressive, this nearest minimum can correspond to a stable representation that is very close to the unseen cue itself rather than a memorized pattern. In other words, the network not only memorizes but also generalizes: the learned basic memories shape the energy landscape so that unseen inputs are mapped to stable attractors capturing their distinctive features.
> > > >
> > > > For example, in Figure 3c, when two unseen images of the digit "6" are presented, the network converges to two distinct attractors that preserve the distinguishing details of each input while still sharing overlapping components in the hidden layer that identify them as class "6". This illustrates an advantage of our model: it supports both memorization and generalization through its learned basic components. We added this discussion to the paper (lines 360-370).
> > > >
> > > > We sincerely appreciate your decision to raise your score and your support for the scientific direction of this work.

---

> > > > > ### Comment · Reviewer_1mum · 2025-11-24
> > > > >
> > > > > Thank you for the edits made. Just some final, scattered thoughts on the generalization capability of your model - broadly speaking, I suppose you could say that this capability is a result of the two-layered structure of your network; the latent layer enabled the network to learn generalizable features (basic ones, in your word), which is impossible for single-layer models like vanilla Hopfield Network. I also see some opportunities in discussing the biological plausibility of the learning rule (e.g., whether it's Hebbian etc.) but that's indeed beyond this paper's scope.
> > > > >
> > > > > Anyways, this paper now reads much more nicely. Raised my score :)

---

> > > > > > ### Author Response · Authors · 2025-11-24
> > > > > > **Authors Response to the Reviewer 1mum - Part 4**
> > > > > >
> > > > > > I appreciate your comments regarding the model’s generalization capability. Your interpretation aligns well with our model, and we will consider adding a brief clarifying remark about the role of the latent layer in enabling more abstract feature learning.
> > > > > >
> > > > > > We also appreciate your suggestion about discussing the biological plausibility of the learning rule. While it is indeed outside the current scope, this is an excellent direction for future work, and we will keep it in mind for subsequent studies.
> > > > > >
> > > > > > Thank you again for your constructive input and for helping improve the clarity and quality of the paper.

---

> > > > > > > ### Author Response · Authors · 2025-11-28
> > > > > > > **Authors Response to the Reviewer 1mum - Part 5**
> > > > > > >
> > > > > > > Regarding your final remark, we have added a paragraph to the Conclusion (lines 496–504) clarifying the role of the latent layer in the model’s generalization capability.
> > > > > > >
> > > > > > > Thank you again for your constructive input and for supporting this work.

---

### Official Review · Reviewer_81Kw · 2025-10-29

**Soundness:** 3
**Presentation:** 2
**Contribution:** 2
**Rating:** 6
**Confidence:** 3

**Summary:**

This work assess a biologically plausible two-layer associative memory network that overcomes the linear capacity constraint of previous models. The core innovation is replacing the winner-takes-all nonlinearity with a threshold activation function, which enables distributed representations where single hidden neurons encode components shared across many memories. Consequently, the network achieves an exponential storage capacity of $2^{N_h}$, supports compositional storage of complex patterns, and demonstrates robust recall from noise on datasets like MNIST and CIFAR-10.

**Strengths:**

1. The network is shown to have exponentially higher capacity than the prior two-layer implementations.
2. The model's fixed points possess large basins of attraction, making the memory retrieval process robust to substantial noise in the visible inputs.

**Weaknesses:**

1. While I certainly agree pairwise interactions between neurons is biological, I do not agree with the implication that setwise interactions are not. I think the claim of this point should be more nuanced, e.g., see appendix A.2 of Burns & Fukai (2023).
2. The theoretical exponential capacity relies on the condition that the number of visible units ($N_v$) is much larger than the number of hidden units ($N_h$), i.e., $N_v \gg N_h$ (see L188-193). This does not seem obviously "biological". It would help if the authors could reference some related neurobiological literature on this point.

**Questions:**

1. The theoretical capacity holds strongest in the limit $N_v \gg N_h$, but the authors achieved successful recall on CIFAR-10 with a ratio of $N_v/N_h \approx 2$ ($1024$ visible vs. $500$ hidden units) (see L346-355). What is the rigorous theoretical lower bound on the ratio $N_v/N_h$ that is necessary to maintain near-exponential capacity and robust recall, especially when dealing with real-world, highly correlated memories?

---

> ### Author Response · Authors · 2025-11-21
> **Authors Response to the Reviewer 81kw**
>
> We thank the reviewer for their valuable time and for providing instructive feedback that has helped us improve our work. In the following, we address all questions and concerns, and we have updated the manuscript accordingly. The revised version is now ready for review.
>
> We would be grateful if the reviewer could consider increasing their score should they find the updated work satisfactory.
>
>
> ### W1. Regarding the set-wise connections:
>
> We thank the reviewer for highlighting this point. We agree that setwise (higher-order) interactions can occur in biological circuits under certain mechanisms. In the revised manuscript, we have softened the language in the introduction: rather than stating that such interactions are “biologically implausible,” we now describe them as “challenging to implement broadly in biological circuits”.
>
> Specifically, we now reference Burns and Fukai (2023, Appendix A.2), which highlights several biological substrates for setwise interactions. We emphasize that while these mechanisms demonstrate that setwise interactions can exist, their restricted nature motivates the use of two-layer architectures (Krotov and Hopfield, 2021) that is based on only standard synapse.
>
> We believe this revision provides a more nuanced and accurate description of the biological evidence while maintaining the motivation for our modeling approach.
>
>
> ### W2. Regarding the biological plausibility of the ratio $\frac{N_v}{N_h}$:
>
> This is indeed a very important point. As shown in Appendix D, even when $N_v = 10 N_h$, exponentially many fixed points remain stable. Thus, this condition is not biologically difficult to satisfy, and future work may help identify the existence of such circuits in the brain. One possible realization is to interpret visible neurons as cortical neurons and hidden neurons as hippocampal neurons, as discussed in Shin et al., Science (2021).
>
> Please also see our response to Q1.
>
> ### Q1. Regarding the ratio $\frac{N_v}{N_h}$ for CIFAR-10 :
>
> This is a very important question. We added Appendix D, clarifying that the probability of correct recall is
> $$
> P_\text{no flip} = \Phi(1/2\sigma_z)^{N_h},
> $$
> where $\Phi$ is the cumulative distribution function of the standard normal distribution. This probability rapidly approaches 1 as the ratio $N_h/N_v$, and consequently $\sigma_z$, becomes smaller (please see Eq.34 in Appendix D). As we show in Appendix D, even when $N_v = 10 N_h$, exponentially many fixed points remain stable.
>
> To determine the number of hidden neurons required for real-world memories with diverse statistics, note that memories from an $N_v$-dimensional space are recalled within at most an $N_h$-dimensional subspace spanned by the $N_h$ basis memories defined by the hidden-to-visible weights. If this subspace is not sufficiently expressive, the reconstructed images will not be recognizable, particularly for complex datasets such as CIFAR10.
>
> In summary, the number of hidden neurons must be small enough relative to the number of visible neurons to guarantee stable recall, yet large enough to capture the statistical structure of the stored memories so that reconstructions remain recognizable.
>
> Finally, to address the ratio $N_h / N_v$ for CIFAR10, we updated our experiments to use the full RGB images of dimension $3072$, while keeping $N_h = 500$, reducing the ratio to approximately $1/6$. Appendix D shows examples of recalled representations for CIFAR10 and MNIST when the number of hidden neurons is insufficient.
>
> Once again, we thank the reviewer for their time and valuable feedback, which greatly helped us improve the clarity and presentation of our work. We hope that our responses have fully addressed all of their questions and concerns.

---

### Official Review · Reviewer_Gxnx · 2025-10-31

**Soundness:** 4
**Presentation:** 4
**Contribution:** 3
**Rating:** 6
**Confidence:** 3

**Summary:**

Associative memory models often struggle with a trade-off between biological plausibility and storage capacity. Previous biologically plausible two-layer models, such as Krotov and Hopfield (2021), were limited by winner-takes-all dynamics, resulting in a storage capacity that scaled only linearly with the number of hidden neurons.
This paper proposes a modification to this two-layer architecture: replacing the winner-takes-all nonlinearity with a simple threshold Heaviside activation function. The authors claim this allows for distributed, compositional representations in the hidden layer, rather than localized representations that tie memory to neurons.
The primary contribution is the theoretical and empirical demonstration that this network achieves an exponential storage capacity, $2^{N_h}$, in the number of hidden neurons ($N_h$), provided the number of visible neurons ($N_v$) is much larger than the hidden neurons ($N_v \gg N_h$).

**Strengths:**

The paper's main idea is simple and effective. It directly addresses a key limitation of the Krotov and Hopfield (2021) model—linear capacity—by changing the activation function. It also provides a simpler alternative to other recent work that required combining multiple modules to achieve exponential capacity.

The theoretical analysis is also nice. The paper provides a clear derivation for how the $N_v \gg N_h$ regime leads to a decoupled hidden layer where all $2^{N_h}$ binary states can be stable fixed points. This theory is backed by numerical experiments. The stability of these points is also analyzed. The experiments on MNIST and CIFAR-10 show that the model can store tens of thousands of highly correlated, real-world patterns. The assumptions made are mostly in line with derivations in related literature and so are the experiments.

The paper is written clearly. It identifies the limitations of prior work and effectively explains how the proposed mechanism overcomes them.

**Weaknesses:**

In the CIFAR-10 experiment, the assumption used to derive the theories, $N_v \gg N_h$, seems to not to be the case as $N_v=1024, N_h=500$, it would be nice to discuss more about the implications of this.

Although not a fault of the paper per se but rather symptomatic of the computational neuroscience literature at large, biological plausibility is a loaded expression and can mean very different things. Also, although the architecture (pairwise synapses, two layers) avoids the implausible interactions of other models , the learning rule is a standard gradient-based optimization of a global cost function, which, despite various research on the biological basis of backprop, remains unclear whether it is truly biologically plausible.

**Questions:**

1, In your classification results table, the nonlinear classifier on original MNIST images achieves 100% accuracy, but on the recalled visible patterns, it only achieves 91%, can you briefly explain this performance drop?

2, What values did the learned threshold $\theta$ converge to for the MNIST and CIFAR-10 experiments?

3, Does the CIFAR-10 results imply the $N_v \gg N_h$ condition is sufficient but not necessary, or is a different mechanism (perhaps from the learning rule) responsible for the high capacity in this regime?

---

> ### Author Response · Authors · 2025-11-21
> **Authors Response to the Reviewer Gxnx**
>
> We thank the reviewer for their valuable time and for providing instructive feedback that has helped us improve our work. In the following, we address all questions and concerns, and we have updated the manuscript accordingly. The revised version is now ready for review.
>
> We would be grateful if the reviewer could consider increasing their score should they find the updated work satisfactory.
>
> ### W1. and Q3. Regarding the ratio $\frac{N_v}{N_h}$:
>
> This is an important observation. We added Appendix D, clarifying that the probability of correct recall is
> $$
> P_\text{no flip} = \Phi(1/2\sigma_z)^{N_h},
> $$
> where $\Phi$ is the cumulative distribution function of the standard normal distribution. This probability rapidly approaches 1 as the ratio $N_h/N_v$, and consequently $\sigma_z$, becomes smaller (please see Eq.34 in Appendix D). As we show in Appendix D, even when $N_v = 10 N_h$, exponentially many fixed points remain stable.
>
> To determine the number of hidden neurons required for real-world memories with diverse statistics, note that memories from an $N_v$-dimensional space are recalled within at most an $N_h$-dimensional subspace spanned by the $N_h$ basic memories defined by the hidden-to-visible weights. If this subspace is not sufficiently expressive, the reconstructed images will not be recognizable, particularly for complex datasets such as CIFAR10.
>
> In summary, the number of hidden neurons must be small enough relative to the number of visible neurons to guarantee stable recall, yet large enough to capture the statistical structure of the stored memories so that reconstructions remain recognizable.
>
> Finally, to address the ratio $N_h / N_v$ for CIFAR10, we updated our experiments to use the full RGB images of dimension $3072$, while keeping $N_h = 500$, reducing the ratio to approximately $1/6$. Appendix D shows examples of recalled representations for CIFAR10 and MNIST when the number of hidden neurons is insufficient.
>
>
> ### W2. Regarding the biological plausibility of the learning rule:
>
> We fully acknowledge that the current learning rule, which relies on backpropagation, is not biologically plausible. Several studies, including the review by Whittington and Bogacz (2019), discuss biologically plausible approximations of error backpropagation in the brain. In future work, we hope to develop and provide a biologically plausible implementation of the learning rule, and we have added this discussion to the conclusion of the paper lines 481-483.
>
> ### Q1. Regarding the classification accuracy:
>
> The original table may have caused some confusion. We used both linear and nonlinear classifiers: the nonlinear classifier achieved $100\\%$ accuracy on both the original MNIST images and the recalled visible patterns, while the linear classifier, being less expressive, achieved $91\\%$.
>
> The previously reported numbers corresponded to the training accuracy on the stored memories. To provide a more reliable assessment of recall quality, we have updated the classification-accuracy table. In particular, we now cue the network with "unseen" test images and examine the recalled representations. We then measure test-set classification accuracy, defined as how accurately the true label of the recalled representation for an unseen image can be predicted by a classifier trained only on the stored images, to evaluate how class-discriminative these recalled patterns remain for unfamiliar cues.
>
> Figure 3c, Figure 5c, and Table 1 have been updated accordingly, along with a clearer explanation of these results.
>
> ### Q2. Regarding the learned thresholds :
> The network learns an effective threshold of $\theta = 0.21$ for the MNIST dataset and $\theta = 0.43$ for the CIFAR-10 dataset. Note that the statistics of the learned basic memories in MNIST and CIFAR-10 differ from one another and from the normal distribution assumed in the theory, which explains the difference between the learned thresholds.
>
>
> Once again, we thank the reviewer for their time and valuable feedback, which greatly helped us improve the clarity and presentation of our work. We hope that our responses have fully addressed all of their questions and concerns.

---

> > ### Comment · Reviewer_Gxnx · 2025-11-26
> >
> > I'd like to thank the author for their timely response. The responses to W1/Q3/W2/Q2 are helpful as well as the updates to the experiments/figures. And the reponse to Q1 is particular interesting to me since, in my understanding, the authors' model learns "basic memories" (components) in the neurons of the hidden layer. These components can be combined to form distributed representations, allowing the network to generalize to "unseen" cues by creating new stable fixed points that capture the composition of the input rather than snapping to a training example. Although it seems intuitively unclear whether this is a bug or feature for an associative memory model, it is surely interesting and useful behavior.
> >
> > In particular, I think it would be appropriate to add a brief discussion in passing with a closely related work in biologically plausible associative memory/representation learning models [1] (in the case of [1] a hierarhical predictive coding model but they compared with dense associative memory as well). [1] discusss in depth what "unseen" could mean---an unseen image of the same semantic content (such as different pictures of mnist "6") is different from an unseen image of a cifar10 image---and discusses what biologically plausible associative memory models can do in such situations. [1] also nicely complements the authors' submission in that they show features can be composed hierarchically to form more semantic features (e.g., edges forming the semantic digit "6"), which seems to have a degree of biological resemblance as well.
> >
> > What seems interesting to me is that these works (current submission, [1], and [2]) converge on the insight that for a biological memory model to handle the statistics of natural images (correlated and complex data), it must learn the underlying compositional structure of that data. A discussion of these parallel findings would provide valuable context on how biologically plausible models are evolving to merge storage with generative understanding.
> >
> > [1] https://direct.mit.edu/neco/article/37/8/1373/131383/Predictive-Coding-Model-Detects-Novelty-on
> > [2] https://arxiv.org/abs/2109.08063

---

> > > ### Author Response · Authors · 2025-11-27
> > > **Authors Response to the Reviewer Gxnx**
> > >
> > > We gratefully acknowledge your thoughtful feedback and the time you invested in re-examining our manuscript. Your comments were highly constructive, and we are pleased that you found our earlier revisions and clarifications satisfactory.
> > >
> > > In response to your valuable suggestion, we have added a discussion connecting our results to related work in hierarchical predictive coding models. This new paragraph appears in the Conclusion (lines 496-504). Thank you for pointing us toward this relevant connection.
> > >
> > > If you feel that the revisions satisfactorily address the concerns, we would greatly appreciate it if you would consider raising your score for our work.

---

> > > > ### Comment · Reviewer_Gxnx · 2025-11-27
> > > >
> > > > I appreciate authors' timely response and updates to the experiments to make their arguments clearer. As some final remarks, I agree with reviewer 1mum that hierarchy could be important for such abstraction/generalization presented in the paper, which could be the motivation to pursue "biological plausibility" (whatever that means) in the first place.
> > > >
> > > > I believe this paper is this a valuable contribution to the associative memory community for its capacity and interesting behaviours. I have raised my score.

---

> > > > > ### Author Response · Authors · 2025-11-28
> > > > > **Authors Response to the Reviewer Gxnx**
> > > > >
> > > > > Examining whether a hierarchical structure is necessary for a biological associative memory network to also generalize is indeed an interesting question, which we will consider in future work.
> > > > >
> > > > > Thank you again for your constructive input, and for your time and engagement.

---

### Author Response · Authors · 2025-12-01
**Rebuttal Summary**

Reviewers requested several clarifications, all of which we addressed. Below is a summary of the main changes.

## Responses to Reviewer Gxnx:
a) To examine sensitivity to the visible/hidden neuron ratio, we added Appendix D showing that robust recall persists even with a ratio as low as 10:1. We also updated the CIFAR10 experiments to use full RGB images instead of grayscale, reducing this ratio.

b) Regarding classification accuracy, we added new experiments and updated Figures3c and 5c as well as Table 1. We now evaluate test-set classification accuracy by cueing the network with unseen images and assessing how class-discriminative the recalled representations remain.

c) we added a new paragraph in the Conclusion connecting our results to hierarchical predictive coding models.

*These additions fully satisfied Reviewer Gxnx, who increased their score to **8**.*

## Responses to Reviewer 1mum:

a) To highlight the novelty relative to Krotov and Hopfield (2021), we added experiments in Appendix C showing that the nonlinearities used by Krotov and Hopfield yield only linear capacity, whereas our model achieves exponential capacity.

b) To assess biological plausibility, we added experiments in Appendix E demonstrating robust recall under asymmetric weights and heterogeneous thresholds.

c) To test the sensitivity of recall to the nonlinearity, we added Appendix F showing that robust recall persists with smooth nonlinearities beyond the threshold function.

d) To examine the sensitivity of recall to the visible/hidden neuron ratio, we did the same as (a) in response to Reviewer Gxnx.

e) To address the existence of minima beyond the training samples, we added new experiments to Figures 3c and 5c demonstrating the existence of exponentially many fixed points that are also structured, enabling both memorization and generalization.

f) To test the sensitivity of recall to time constants, we added Appendix G showing that exponentially many fixed points remain stable even when visible neurons are only two times slower than hidden ones.

g) we added a new section discussing the biological plausibility of the model.

h) we revised the structure of sections in the paper for improved clarity.

i) we added a discussion of the model’s generalization capabilities.

*These additions fully satisfied Reviewer 1mum, who increased their score to **6**.*

## Responses to Reviewer 81kw:
a) Regarding setwise (higher-order) connections, we revised the introduction to refine the language and describing such interactions as “challenging to implement broadly in biological circuits” rather than “biologically implausible.” We also added a citation to Burns and Fukai (2023), which documents biological substrates for setwise interactions, while emphasizing why two-layer architectures remain well-motivated.

b) We clarified, by adding new experiments in Appendix D, that exponentially many fixed points remain stable even when $N_v = 10 N_h$, and we suggested a biological interpretation mapping visible neurons to cortical neurons and hidden neurons to hippocampal neurons (Shin et al., 2021).

c) To examine the sensitivity of recall to the visible/hidden neuron ratio, we did the same as (a) in response to Reviewer Gxnx.

*Reviewer 81kw likely did not have the opportunity to engage in the rebuttal, so their score remained at **6**.*

**Reviewers recognized the paper as a valuable contribution, introducing a biologically plausible associative memory with exponential capacity, compared to the state-of-the-art and promising biologically plausible model of Krotov and Hopfield (2021), which has only linear capacity.**

---

### Meta-Review · Area_Chair_sNRD · 2026-01-12

**Summary:**

Reviewers’ main hesitation wasn’t about whether the core idea is interesting (most agreed it is), but about how strongly the paper can claim novelty + biological plausibility, and how cleanly the theory matches the experimental regime.

These were partly mitigated by the rebuttal (new appendices adding comparisons, heterogeneity/asymmetry tests, smoothing robustness, and ratio/time-constant analyses) , which is why I still lean accept **but only if the authors treat the biological framing more cautiously and make the theory-to-experiment mapping explicit**.

**Reviewer Concerns:**

Addressed (some partially)

- explicit comparisons with 'Krotov & Hopfield'.

- validation that the theoretical step-function story is not brittle under smoothing: the rebuttal adds robustness checks under smoothed nonlinearities.

- biological realism: asymmetric weights, heterogenous thresholds, and related deviations from the idealized setup
20339-Reviews and Form

- additional ratio / time-constant analysis and supporting experiments, which meaningfully improves the theory-to-practice bridge.

- softened/qualified parts of the biological discussion (including setwise interactions language).

Still outstanding:

- Biological plausibility is still not fully convincing at the learning-rule level.
- Even with architectural plausibility improvements, the training is still described as 'standard gradient-based optimization of a global cost function', which is exactly the core plausibility objection
- Novelty vs the broader “modern Hopfield / dense associative memory” landscape remains incomplete.
The new comparisons help, but the rebuttal still doesn’t fully situate the method against the broader set of contemporary associative memory models (especially in terms of practical regimes and empirical head-to-heads).
- Practical operating regime and tradeoffs are still under-specified for real-world use.
The rebuttal adds ratio/time-constant analysis, but the paper still needs a cleaner message

**Reviewer Scores:**

- 1mum from 2 to 6
- the other two remain at 6, 6

---

### Decision · Program_Chairs · 2026-01-26

Accept (Poster)